# Polyelectrolyte Multilayer Films Based on Natural Polymers: From Fundamentals to Bio-Applications

**DOI:** 10.3390/polym13142254

**Published:** 2021-07-09

**Authors:** Miryam Criado-Gonzalez, Carmen Mijangos, Rebeca Hernández

**Affiliations:** Instituto de Ciencia y Tecnología de Polímeros, CSIC, Juan de la Cierva 3, 28006 Madrid, Spain; cmijangos@ictp.csic.es

**Keywords:** natural polymers, polysaccharides, polyelectrolytes, layer-by-layer, electrostatic interactions, biomedical applications

## Abstract

Natural polymers are of great interest in the biomedical field due to their intrinsic properties such as biodegradability, biocompatibility, and non-toxicity. Layer-by-layer (LbL) assembly of natural polymers is a versatile, simple, efficient, reproducible, and flexible bottom-up technique for the development of nanostructured materials in a controlled manner. The multiple morphological and structural advantages of LbL compared to traditional coating methods (i.e., precise control over the thickness and compositions at the nanoscale, simplicity, versatility, suitability, and flexibility to coat surfaces with irregular shapes and sizes), make LbL one of the most useful techniques for building up advanced multilayer polymer structures for application in several fields, e.g., biomedicine, energy, and optics. This review article collects the main advances concerning multilayer assembly of natural polymers employing the most used LbL techniques (i.e., dipping, spray, and spin coating) leading to multilayer polymer structures and the influence of several variables (i.e., pH, molar mass, and method of preparation) in this LbL assembly process. Finally, the employment of these multilayer biopolymer films as platforms for tissue engineering, drug delivery, and thermal therapies will be discussed.

## 1. Introduction

Over the last decades, polymers obtained from biomass, known as natural polymers, have been widely employed for a variety of biomedical applications due to their intrinsic properties such as biocompatibility and non-toxicity. They are naturally produced in large quantities because they are present in or created by living organisms such as plants, microorganisms, algae, and animals [1]. Polysaccharides, proteins, and polyamino acids are the most employed natural polymers for biomedical applications. Polysaccharides are constituted by monosaccharide units linked by O-glycosidic bonds including cellulose, chitin, chitosan, alginate, hyaluronic acid, chondroitin sulphate, dextran, and agarose [2]. Proteins, such as collagen and gelatin, are polymer structures formed by 20 different amino acids linked by amide (or peptide) bonds [3]. With regard to polyamino acids, a small group of polyamides consisting of only one type of amino acid linked by amide bonds, poly(L-lysine) (PLL), poly(g-glutamic acid) (PGA), and polyarginyl–polyaspartic acid are extensively employed for biomedical applications [4].

Natural polymers are biodegradable under physiological conditions through enzymatic degradation or hydrolytic processes leading to body-friendly degradation products. However, rapid degradation can be undesirable for their employment in some biomedical applications (i.e., controlled drug release or tissue engineering) [5]. In order to modulate their degradation rate, natural polymers are often chemically modified or cross-linked to yield more stable polymers under physiological conditions. However, this can give rise to toxic by-products or even induce toxic effects of the modified material [6]. A strategy to increase the biostability of natural polymers in physiological conditions without compromising their biocompatibility is the formation of polyelectrolyte complexes (PECs) formed by electrostatic interactions between natural polycations and polyanions. In this regard, layer-by-layer (LbL) assembly, based on the sequential deposition of interacting species onto a substrate, emerged as a versatile, simple, efficient, reproducible, and flexible bottom-up technique, that allows a precise control over the thickness and compositions at the nanoscale [7]. Altogether, this makes LbL assembly one of the most useful techniques for building up advanced multilayer polymer structures and implant coatings towards multiple applications in diverse fields such as biomedicine [8,9], energy [10,11], optics [12,13], or coatings [14,15].

In 1999, Elbert et al. [16] developed, for the first time, a procedure to coat biological surfaces (i.e., a proteinaceous surface) with thin polymer layers of two natural polymers, PLL and alginate (ALG), through LbL assembly in physiological conditions. Since then, different multilayer systems based on natural polymers have been studied, mostly comprising four different polycations, PLL, chitosan (CHI), collagen (COL), and gelatin (GL), and a diversity of polyanions, hyaluronic acid (HA), ALG, chondroitin sulphate (CHS), etc., and different LbL techniques, i.e., dipping, spray, spin coating, or brushing. It has been discovered that the electrostatic assembly of multilayer structures through LbL assembly can be influenced by different parameters such as pH [17], temperature [18,19,20], solvent [21,22], ionic strength [23,24,25] and type and properties of every polyelectrolyte, leading to multilayer films with tailor-made properties for specific applications mainly in the biomedical field.

This review article provides an overview of the state-of-the-art concerning layer-by-layer assembly of natural polymers, from fundamentals regarding the influence of different experimental parameters on the final morphology and properties of the resulting natural multilayers to the biomedical applications. Firstly, the main LbL deposition techniques employed for the buildup of natural polymers leading to multilayer structures and coatings will be addressed, highlighting the influence of different parameters, such as molecular weight or the nature of the polyelectrolytes, on the self-assembly process and architecture of the resulting films. Then, different strategies for the design of natural multilayer films for biomedical applications will be described including procedures for cross-linking or tunability of the architecture of the multilayers to modulate cell adhesion. Finally, the employment of multilayers derived from natural polymers as bioactive functional coatings and platforms for tissue engineering or drug delivery and thermal therapies will be discussed.

## 2. Growth Mechanisms in LbL Assembly of Natural Polymers

Multilayer assembly through the LbL technique involves different types of intermolecular interactions, e.g., electrostatic, charge-transfer, host–guest, hydrophobic interactions, coordination chemistry interactions, and hydrogen and covalent bonding being the electrostatic interactions that are the most studied [26]. The electrostatic interaction takes place between molecules and surfaces that are electrically charged. LbL assembly based on electrostatic interactions gives rise to multilayer films with a well-controlled structure, composition, and thickness by alternate deposition of oppositely charged molecules. In this regard, many natural polymers are polyelectrolytes; that is, they bear ionizable groups within their chemical structure and, therefore, their assembly through the LbL technique occurs through electrostatic interactions between a positively charged polymer (polycation) and a negatively charged polymer (polyanion) (Figure 1a) [27]. Among different LbL techniques, the most employed are dipping, spraying, and spin coating. Dipping LbL is a time-consuming process that consists of immersing a substrate alternately into aqueous polycation and polyanion solutions with a washing step between deposited layers, allowing them to cover substrates of almost any shape and size (Figure 1b) [27,28]. On the other hand, spray-assisted LBL is a time-saving technique based on the sequential spraying of polycation and polyanion solutions leading to multilayer films in few seconds, which becomes attractive for automatization and can scale-up to the industrial level (Figure 1c) [29,30]. Spin LbL assembly is also a time-saving technique, and it is based on spinning a substrate to facilitate the deposition of polymers (Figure 1d) [31,32]. Recently, a new LbL technique, named brush LbL, was developed by Park et al. [33], and it is based on the sequential brushing of polyelectrolyte solutions over a substrate (Figure 1e). They identified brushing LbL as a reliable and more efficient multilayer film-construction process compared to conventional LbL methods for practical applications in dental and clinical situations.

Two main buildup mechanisms, linear and exponential, have been reported during the electrostatic LbL assembly. Linear growth is the simplest one in which the thickness and mass of the film increase linearly with the number of deposited bilayers, and each polyelectrolyte layer interpenetrates only with the adjacent ones [27]. In contrast, in the case of exponential growth, there is a high chain mobility in the direction perpendicular to the film and in the plane of the film leading to the diffusion of at least one of the polymer constituents in and out of the film architecture with the subsequent exponential thickness increase with the number of deposited bilayers [34]. The growth mechanism of multilayer films prepared from PLL as polycation has been reported to be exponential [16,35]. The assembly process of multilayer films obtained by interaction of PLL with HA is characterized by two growth regimes. At the beginning of the deposition process, the surface of the substrate is covered by isolated islands that grow by the deposition of more polymer layers on top and by mutual coalescence until a continuous film is obtained, and after approximately eight deposited bilayers, it displays linear growth. At this point, the second regime begins, showing exponential growth as the number of deposited bilayers increase due to the diffusion of free PLL chains ‘into’ whole film when the film is in contact with a PLL solution and ‘out’ of the film when the film is further brought in contact with an HA solution interacting with HA chains at the outer limit of the multilayer [36,37,38]. Nevertheless, when PLL interacts with PGA, the exponential growth was not only attributed to the diffusion of PLL chains into the whole structure [35] but also to the diffusion of PGA chains into the whole film [39]. As in the case of poly(L-lysine), when CHI is assembled with HA, initially the surface of the substrate was covered by isolated islets that grew and coalesced as the number of deposited bilayers increased until a continuous film was obtained [40]. This process gave rise to an exponential growth due to the diffusion of CHI molecules within the film, which has also been observed when CHI was assembled with other polyanions such as dextran sulfate (Dex) and heparin (HEP) [24,40,41]. In all these cases, CHI/HA, CHI/Dex, and CHI/HEP, CHI is the dominating species of the two polymers, and electrostatic interactions are accompanied by other short-range interactions such as hydrogen bonding [42]. In contrast to the behavior observed for PLL and CHI, the assembly of COL with HA exhibited a linear growth, and it was proven that COL did not diffuse into the film and interacted only with its outer layer. However, the films were not constituted of homogeneously distributed polyanion/polycation complexes, but they were formed of fibers whose width increased with the number of deposition steps [43]. The fibrillary structure of the layers was also observed when COL was assembled with CHS and HEP [44].

The effect of the method of preparation in the buildup process was firstly studied by Porcel et al. [34], who built up PLL/HA films via spray-assisted LbL and dipping LbL. In both cases, the film growth first evolved exponentially with the number of deposited bilayers and, after a given number of deposition steps, its thickness evolution became linear again. This second transition was investigated in detail through spray LbL, reaching the conclusion that this transition always took place after approximately 12 deposition steps independently of the parameters controlling the deposition process, time of spraying, and polyelectrolyte concentrations. The multilayer assembly of CHI and ALG by dipping LbL up to 200 bilayers gave rise to a linear growth showing the final film a thickness of ~35 µm (Figure 2b). However, the layer-by-layer assembly of these two polyelectrolytes, CHI and ALG, by spray-assisted LbL exhibited a totally different behavior. In that case, the thickness increased linearly up to five bilayers, to be turned exponential from 5 to 20 deposited bilayers due to the diffusion of chitosan ‘in’ and ‘out’ of the film during the deposition process, probably to the fast self-assembly process of spray LbL compared to dipping LbL, which makes possible a different arrangement of the polymer chains (Figure 2c). Similar findings were discovered by Decher and co-workers during the LbL assembly of chitosan and carboxymethylchitosan by dipping and spray to obtain highly hydrated films, where the polyanion and polycation possess the same polymer backbone. Multilayer films built up by dipping showed a linear growth with a thickness of ~150 nm for (CMCHI/CHI)_10_, whereas a lower thickness was achieved during spray LbL (~90 nm) but exhibiting and exponential growth [45]. They also studied the influence of the nozzle spray angle on the LbL deposition of chitosan and cellulose nanofibrils (aCNF). Spraying at 90° gives rise to films with homogeneous in-plane orientation, whereas spraying at angles lower than 90° leads to in-plane anisotropic films [46]. Apart from those methods, CHI/ALG films can be built up by brushing LbL leading to nanometer scale thickness. The brushing process follows the same linear growth mechanism as dipping LbL and the resulting films showed similar surface roughnesses (R_q_), R_q_ = 5.7 nm and 5.6 nm for films prepared by dipping and brushing, respectively (Figure 2d) [33].

## 3. Experimental Factors Influencing the Growth in LbL Assembly of Natural Polymers

Even though the same general characteristics can be found regarding the growth mechanisms, the growth process is influenced by a series of parameters that are intrinsic to the polyelectrolytes involved within the LbL buildup such as molecular weight or nature of the polyanion and others specific to the experimental parameters such as pH, ionic strength, or the solvent employed. In this section, the experimental factors influencing the LbL growth process of PLL and CHI acting as polycations will be discussed since they are the most studied in the literature.

### 3.1. Molecular Weight

In the case of multilayer films based on PLL, the film thickness increased after each deposition step, independently of the molecular weight in the linear growth regime. Otherwise, it had a significant influence on the exponential regime. Low molecular weights allowed for the PLL chains to diffuse into the whole film during each deposition step, whereas a high molecular weight restricted the PLL chain’s diffusion into the upper part of the film [49]. The effect of the CHI’s molecular weight on the thickness was also evaluated. For a constant molecular weight of the polyanion (~400,000 Da) and molecular weights of chitosan of 30,000 and 160,000 Da, at a constant pH and ionic strength, it was proven that higher molecular weights gave rise to higher thicknesses; however, the exponential growth rate was the same for high and low CHI molecular weights [50]. When the molecular weight of CHI increased up to 460,000 Da, it was observed that the tendency was the opposite, and the exponential growth was faster for a molecular weight of chitosan of 110,000 Da than for 460,000 Da [40]. The surface morphology was also affected by the molecular weight. High molecular weights of CHI gave rise to a shorter island growth and coalescence stage as well as an earlier transition from islands to a vermiculate morphology than low molecular weights [50]. Decher and co-workers modified chitosan with carboxymethyl groups to obtain zwitterionic or anionic chitosan derivatives, allowing the LbL assembly of multilayer films in which both the polyanion and polycation possess the same polymer backbone, named as matched chemistries. They studied the influence of the molar mass of the polycation, by using low (LMW ~33,000 Da) and medium mass (MMW ~115,000 Da) chitosan on this LbL assembling process and keeping fixed the molar mass of the polyanion (CMCHI ~20,000 Da). It was observed that the stronger complexation occurred between polyanions and polycations of different (‘nonmatching’) lengths (Figure 3a). This can be due to the fact that chain–chain interactions in polyelectrolyte complexes are stronger in the case of comparable charge densities because similar chain lengths lead to denser and less hydrated complexes [45].

### 3.2. Nature of the Charged Groups of the Polyanion

The nature of the charged groups of the polyanion also influences the assembly of the multilayer films. Taking into account that electrostatic interactions are important in the film’s buildup, the quantification of internal ion pairing (extrinsic versus intrinsic charges) and water content were studied for three different systems based on poly (L-lysine), PLL/HA, PLL/CHS, and PLL/HEP in order to examine the influence of the COO^—^ and SO_3_^–^ groups on the film growth, the water content, and the ion pairing. Although these polyanions differed in their charge, the disaccharide units attracted approximately two lysine groups per monomer. The percentage of free NH_3_^+^ in the films decreased as the charge density of the disaccharide increased, and it was related to PLL diffusion directly influencing the film’s growth. It was also proven that PLL/HA and PLL/CHS films were the most hydrated. The selective cross-linking of carboxylate and ammonium ions via carbodiimide chemistry allowed for the determination of the COO^-^/NH_3_^+^ and SO_3_^-^/NH_3_^+^ ion pairing, showing that 46% of NH_3_^+^ groups were unpaired in the PLL/HA films, 21% in the PLL/CHS films, and none in the PLL/HEP films, reaching to the conclusion that this ratio was close to the stoichiometry of these groups in the disaccharide monomeric unit (2:1 for PLL/HA films and 1:1 for PLL/CHS films) [53].

Regarding CHI-based films, the influence of three different polyanions, HEP, HA, and CHS, was studied in LbL assembly by dipping. The most hydrophilic films (CHI/HA) were formed by the assembly of a weak polycation (CHI) and a weak polyanion (HA), and the most hydrophobic ones (CHI/HEP and CHI/CHS) were formed by the combination of weak (CHI)–strong (HEP or CHS) polyelectrolytes. The assembly of two weak polyelectrolytes CHI/HA reduced ion pairing and enabled the swelling of the film, whereas the combination weak–strong polyelectrolytes reduced the swelling because of an increase in ion pairing (Figure 3b) [51]. On the other hand, Schaaf and co-workers also studied the influence of three different polyanions, HA, ALG, and CHS, in the simultaneous spray-assisted assembly with CHI. The maximum film growth rate was observed for the HA/CHI, ALG/CHI, and CHS/CHI charge ratios of ~0.7, ~0.9, and ~1.2, respectively. The simultaneous spray process was governed by the formation of complexes at the interface with the largest complex diameter for a polyanion/CHI molar charge ratio around 0.6 for CHI/HA, 1.2 for CHI/ALG, and 1.1 for CHI/CHS (Figure 3c) [52]. Neto et al. demonstrated that the self-assembly of CHI with HA gave rise to an exponential growth, whereas with HA–DN showed a linear growth. It was found that CHI/HA films were rougher than CHI/HA–DN films, leading to a higher surface area and higher mass deposition [54].

### 3.3. pH, Ionic Strength, and Solvent

The driving force of the assembly is influenced by the pH, ionic strength, and solvent employed. Multilayer PLL/PGA films showed different behaviors depending on the pH of the assembly. The main driving force of the assembly at pH 7.4 is the electrostatic interaction, whereas hydrogen bonding and hydrophobic interaction are the dominant interactions in films built up at low pH [55]. The growth of these films was higher at acidic pH [40]. When PLL is assembled with HA, the acid–base equilibria of multilayer PLL/HA films showed that these films can be electrostatically adsorbed under highly charged ‘sticky’ conditions but then quickly transformed into stable low-friction films simply by altering their pK_a_ on adsorption, at the same pH environment [56]. The effect of the pH and ionic strength in the growth process of CHI/Dex and CHI/HEP films showed an increase in the film’s thickness with the increase in the NaCl concentration at a fixed pH [24]. The same effect was found for CHI/HEP films when the pH increased at a fixed ionic strength (Figure 3d) [42,57]. In the case of CHI/HA films, at low salt concentrations (10^−4^ M NaCl), the surface of the substrate was covered by islets with up to 50 bilayers with a linear increase in the film’s growth. At a high salt concentration (0.15 M NaCl), the formation of a uniform film took place only after a few deposition steps showing an exponential growth (Figure 3e) [25]. When CHI was assembled with ALG, the study of the buildup’s process at different concentrations, pH, and ionic strength allowed to conclude that the fastest film growth took place for chitosan and alginate concentrations of 1.0 and 5.0 mg/mL and pH 5 and 3, respectively, conditions under which alginate is in high concentration and only partially ionized in a way that its negative charge interacts weakly with the positively charged amino groups of CHI [47,58]. The effect of the solvent in the assembly process of CHI and PGA showed that the adsorption process from an aqueous phase was not stable; however, the use of an organic solvent as a less soluble solvent gave rise to thicker films and achieved stable deposition [59].

## 4. Engineering of LbL Films for Biomedical Applications

As mentioned before, natural polymers are of great interest in the biomedical field due to the following features: (i) biodegradability: they do not show adverse effects on the environment or human being; (ii) biocompatibility and non-toxicity: almost all of these materials are carbohydrates in nature and composed of repeating monosaccharide units; (iii) economic: they are cheaper than synthetic polymers; (iv) safety: they do not have side effects, whereas synthetic polymers could produce side effects; (v) availability: they are produced naturally in large quantities [4,60]. In this section, we first describe some parameters that highly influence the biological response of multilayer films to be employed in biomedical applications.

### 4.1. Cross-Linking Degree

To be employed for biomedical applications, LbL assembled systems have to be biocompatible and, in parallel, it is necessary to know their behavior at physiological conditions. In this regard, it is important to note that cross-linking of multilayer polymer films influences the biodegradation and, hence, it constitutes an important strategy for the design of biomaterials. There are different kinds of cross-linking processes. The most used are chemical cross-linking, using the carbodiimide chemistry (1-ethyl-3-(3-(dimethylamino)-propyl) carbodiimide (EDC) in combination with N-hydroxy-sulfosuccinimide (NHS)), and thermal cross-linking, employing high temperatures during a time interval which are different for every polymer system. Both cross-linking processes give rise to amide bond formation between carboxylic groups of the PA and amine groups of the PC [61,62]. In the case of polycations with primary amine groups, such as chitosan or collagen, another covalent cross-linking process can take place employing genipin, a natural origin polymer that acts as a cross-linking agent between primary amine groups (Figure 4a) [63,64]. Multilayer PLL/HA films cross-linked using the carbodiimide chemistry showed a high resistance to hyaluronidase, an enzyme that naturally degrades hyaluronan [61]. Cell adhesion of L929 fibroblast cells on CHI/CHS also improved after cross-linking with genipin [65,66]. In the case of CHI/HA films, the degradation evaluated in vitro, in contact with enzymes, plasma, and macrophages, and in vivo, in mouse peritoneal cavity, could be tuned by film cross-linking. Native films showed degradation by enzymes, whereas cross-linked films were more resistant to enzymatic degradation. This effect was also observed for plasma contact which induced changes in the structure of native films but not in cross-linked ones. On the contrary, even if cross-linking can minimize the degradation induced by cells, it cannot inhibit it, as it was studied in vivo. Native films showed an almost complete degradation, whereas cross-linked films were only partially degraded [67]. When COL acts as polycation, non-cross-linked multilayer films COL/CHS are stable in culture media at physiological conditions, whereas COL/HA, COL/HEP and COL/ALG films are unstable [9,44,68]. Nevertheless, the biodegradation can be avoided by chemical cross-linking of these films giving rise to stable membranes [9,68].

Cross-linking of multilayer films not only alters their biodegradation, but it also gives rise to an increase in the film’s rigidity, which, in turn, highly influences cell adhesion and cell proliferation in LbL films, a key parameter for the design of polymer materials for biomedical applications. Multilayer PLL/HA films cross-linked using the carbodiimide chemistry showed an improved cell adhesion, whereas the non-cross-linked native films were highly cell anti-adhesive [61], and this effect was detected for different kind of cells [69,70,71,72,73,74]. As in the case of PLL-based films, the cross-linking degree also influences the cell adhesion on CHI-based films. Chemical cross-linked CHI/HA multilayer films showed an improvement in the mechanical properties with the subsequent increase in human umbilical vein endothelial cells (HUVECs) adhesion, spreading, and proliferation (Figure 4b) [75]. This was not only observed for CHI/HA films but also for CHI/ALG films [70,76,77].

**Figure 4 polymers-13-02254-f004:**
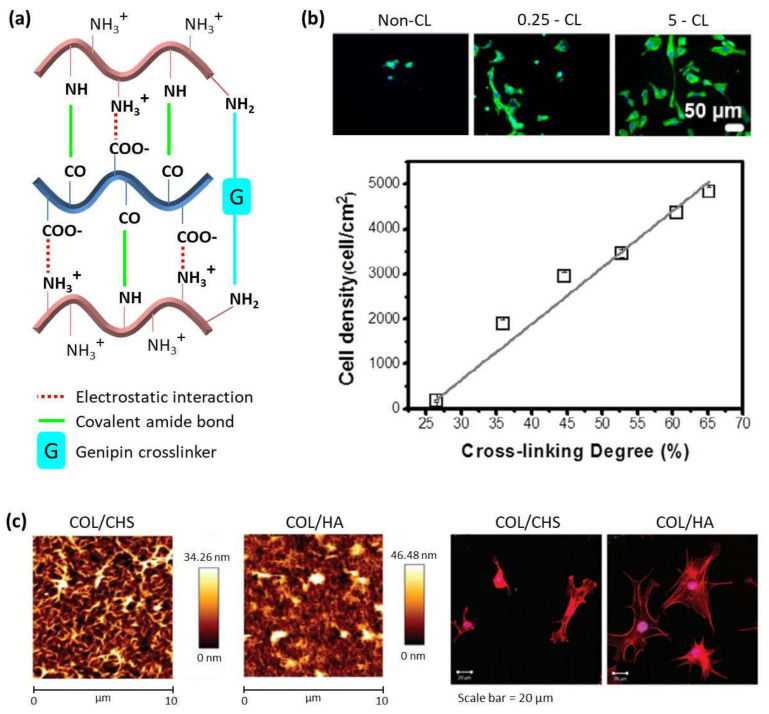
(**a**) Schematic representation of different cross-linking mechanisms in LbL films made of chitosan and alginate or hyaluronic acid; (**b**) DAPI-phalloidin fluorescence assay at 1 day of HUVECs cell culture in multilayers with different cross-linking degrees (top) and cell density as a function of cross-linking degree (bottom). (Adapted and reprinted with permission from [75]; Copyright 2017 Acta Materialia Inc. Published by Elsevier Ltd. All rights reserved.); (**c**) AFM images of (CHI/CHS)_10_ and (CHI/HA)_10_ films (top) and phalloidin-stained bovine chondrocytes cultured on (CHI/CHS)_10_ and (CHI/HA)_10_ films (bottom) (Adapted and reprinted with permission from Ref [44]. Copyright 2011 American Chemical Society).

### 4.2. Structure and Nature of the Ending Layer

Apart from film stiffness and cross-linking degree, the number of deposited bilayers and the nature of the ending layer can also influence cell adhesion. In the case of PLL-based films, cell adhesion was more favorable when HA acted as a counterion compared with CHS [78]. In addition to this, cell adhesion was also enhanced by the presence of growth factors (bFGF) into the multilayer structure of PLL/CHS and PLL/HA films [78]. For CHI-based films, it was also proven that ALG-ending films were more adhesive than the CHI ending ones, and adhesion increased with the number of bilayers as well [47,58]. The incorporation of sulfated levan (L–S) within the multilayer CHI/ALG structure gave rise to a fourfold increase in adhesion strength in comparison to the native films without L–S [79,80]. Moreover, the effect of the modification of HA with dopamine (HA–DN) in the multilayer assembly with CHI was explored and compared with native CHI/HA films, showing that CHI/HA–DN films possessed enhanced cell adhesion, proliferation, and viability [54,81]. However, not only counterions can be modified to tune cell adhesive properties; mesyl and tosyl–CHI (mCHI) were self-assembled with ALG leading to anti-adhesive films in contact with human adipose stem cells (hASCs). Interestingly, even if mCHI/ALG multilayer films were non-cell-adhesive, they were non-toxic toward hACSs, and they could be used as a scaffold for the formation of 3D aggregates of hASCs [82]. The COL/ALG films possess excellent properties for cellular adhesion and proliferation of human umbilical vein endothelial cells (HUVECs) promoting their use as cell-stimulating materials to coat prostheses for in vivo applications such as inner lining of lumens for vascular and tracheal implants [9]. In vitro experiments with diverse kind of cells, such as chondrocytes and chondrosarcoma cells, in different films with collagen acting as polycation, COL/HA, COL/CHS, and COL/HEP, proved that films promoted excellent cell adhesion when COL was the outermost layer providing insight into the use of these multilayer films for biomedical applications (Figure 4c) [43,44]. Cell adhesion is also influenced by the swelling behavior, as it was exhibited on multilayer PLL/PGA films. When films were built-up at basic pH, they shrank in contact with salt-containing solutions, and they were found to be highly cell adhesive, whereas those assembled at an acidic pH swelled being highly cell resistant [40]. The adhesion of cerebral cortical neural stem/progenitor cells (NSPCs) on these films achieved large neural network size and a large number of functional neurons making them accurate for neural regeneration [83].

## 5. Overview of the Performance of Natural LbL Films in Selected Biomedical Applications

Among the different applications for LbL-assembled natural polymer films, the most studied are coatings for surface modification designed to endow substrate biomaterials with additional functionalities (bioactivity, blood-compatibility or antimicrobial response, among others), platforms for tissue regeneration, and matrixes for drug delivery. Recently, LbL nanocomposite films prepared through the addition of inorganic nanoparticles (NPs) (gold or iron NPs, among others) within LbL films have been employed for selected applications including platforms for thermal treatments such as magnetic hyperthermia.

### 5.1. Multilayer Films as Bioactive Functional Coatings

#### 5.1.1. Blood-Compatible Coatings for Cardiovascular Implants

Multilayer films from natural polymers have been employed as blood-compatibility materials for coating of cardiovascular implants. The use of CHI-based films as blood-compatibility materials depends on their thrombogenic potential and the hemocompatibility. The extent of platelet adhesion and surface-induced activation are early indicators of thrombogenic potential, and the clotting time of activated partial thromboplastin time (APTT) together with the prothrombin time (PT) are indicators of the coagulation activation. The anti- vs. procoagulant activity of films based on CHI was dependent on the salt concentration, the number of deposited bilayers, and the nature of the polyanion. The bioactivity of CHI/HEP films was different from CHI/Dex films and the surface of the CHI/HEP assembly showed strong anticoagulant activity which is useful for coating various blood-contacting biomedical materials [24]. Chitosan and N-methylated chitosan derivatives, namely, N,N-dimethyl chitosan (DMC) and N,N,N-trimethyl chitosan (TMC) assembled with HEP, showed good inhibition of natural inflammatory response hindering the proliferation speed of fibroblasts up to a factor of forty-fold [84]. In that sense, multilayer CHI/HEP films have been employed to coat different materials, such as stainless-steel coronary stents and NiTi endovascular stents, to accelerate the re-endothelialization and healing process after coronary stent deployment. In vitro studies proved that these coatings decreased the platelet adhesion in comparison with uncoated NiTi stents, displaying good hemocompatibility and enhanced thromboresistance. In vivo studies in a porcine coronary injury model and arteriovenous shunt model demonstrating that this coating promoted re-endothelialization and good hemocompatibility with improved anticoagulation properties [8,85]. Multilayer CHI/HA films have been used to repair porcine arteries by placing them on damaged arteries. A strong adhesion of the coating on the artery was observed when CHI was the contact surface due to the fact that it is a polycation and exhibits excellent bio-adhesive properties toward negatively charged surfaces presented by damaged arteries [86].

Apart from CHI-based films, COL/HEP films have also been applied to improve the blood compatibility of titanium because they decrease platelet adhesion and activation and prolong the APTT and PT, indicating low coagulation activation. Multilayer titanium-coated COL/HEP structures presented enhanced anticoagulation properties for potential applications such as cardiovascular implants [87,88,89]. These COL/HEP multilayers have also been used to cover stainless-steel coronary stents (Figure 5) [90].

#### 5.1.2. Biocompatible Coatings for Dental and Orthopedic Implants

Besides cardiovascular coatings, CHI/HEP films can be employed to modify the surface of titanium implants, improving biocompatibility for use in dental or orthopedic implants, as it was checked in vitro with osteoblast cells improving their adhesion, proliferation, and differentiation [92]. In this sense, basic fibroblast growth factor (bFGF) was bound to CHI/HEP multilayers to enhance the bone marrow-derived ovine (MSCs) cell proliferation to be employed as promising surface coatings that can stabilize and potentiate the activity of growth factors for therapeutic applications [93]. Multilayer CHI/ALG and CHI/HA films have also been employed as titanium coatings [81,94]. The adhesive character of CHI/HA films was further improved by chemical modification of HA with dopamine (HA–DN). The osteogenic differentiation of mouse MC3T3-E1 pre-osteoblasts cells toward the formation of a mineralized extracellular matrix was studied and (CHI/HA-DN)_200_ films showed enhanced cell adhesion, proliferation, and differentiation. The adhesion properties of the multilayer films (CHI/HA)_200_ and (CHI/HA-DN)_200_ was also evaluated by shear strength of single-lap-joint adhesively bonded metal specimens by tension loading. The results showed an increase in the adhesion strength from 3.4 MPa for HA-based films to 8.6 MPs for HA–DN-based films, highlighting the enhanced adhesive properties of CHI/HA–DN films. Then, the bio-adhesiveness of these free-standing membranes were evaluated in contact with the surface of a piece of porcine bone, observing that it is necessary for a higher force to pull out CHI/HA–DN films than CHI/HA films. All these features make CHI/HA–DN films ideal candidates for bone tissue regeneration [91]. Coating of magnesium-based degradable scaffolds with PLL/ALG films is aimed to improve their surface bioactivity. For that purpose, they were cross-linked using the carbodiimide chemistry and surface functionalized with fibronectin immobilization. The in vitro cytocompatibility studies with MC3T3-E1 osteoblast cells demonstrated that the pretreatment of the magnesium substrate greatly influenced the biocompatibility of the films proving that fluoride pretreatment is necessary for the long-term stability of PLL/ALG films and, therefore, for the slow corrosion of the magnesium substrates in order to be applied as matrixes for the delivery of drugs and other biomolecules for successful bone regeneration in vivo [95].

#### 5.1.3. Antimicrobial Coatings

Prevention of pathogen colonization during surgery is an important medical issue playing an important role in the development of biomaterials. That is the case of medical gauzes which are in direct contact with body fluids and organs during surgery and implanted materials. Regarding gauzes, the incorporation of antimicrobial molecules on textile materials is essential. Gomes et al. [96] coated cotton fibers by alternate deposition of chitosan and alginate layers to evaluate their antibacterial properties in contact with *Staphylococcus aureus* (*S. Aureus*) and *Klebsiella pneumonia*. Results showed a bacteriostatic effect of coated cotton fibers on both bacteria tested for potential application in the health-care field. Apart from that, immobilization of antimicrobial molecules on biomaterials surfaces is of vital importance to prevent bacteria and yeasts employing LbL technique have been developed. Multilayer CHI/HEP films were used to coat poly(ethylene terephthalate) (PET) films in order to control the degree of interpenetration of the layers and the antibacterial properties by altering the assembly pH. The antibacterial property was proved in vitro with *Escherichia coli* (*E. coli*) bacteria, showing that the number of adhered bacteria decreased with a decrease in the assembly pH making these nanostructured films useful as powerful anti-infection coatings for medical devices [97]. The same results were observed when these CHI/HEP films were used to coat polystyrene films [98]. The bacterial resistant properties were also observed for the combination of CHI with other polyanions, such as HA [40]. Multilayer films formed by cateslytin functionalized hyaluronic acid (HA-–CTL), an endogenous host-defensive antimicrobial peptide, and chitosan were developed. Hyaluronidase secreted by the pathogens led to film degradation and subsequent antimicrobial action of cateslytin, inhibiting the growth of Gram-positive *Staphylococcus aureus* bacteria and *Candida albicans* yeasts after 24 h incubation. The antibacterial and antifungal ability of the coating, together with the biocompatibility tests in contact with fibroblasts, allowed to anticipate the employment of HA–CTL/CHI films to prevent infections on catheters or tracheal tubes [99]. The CHI/HA films containing calcium phosphate have been also used as an antibacterial substrate for stromal cell adhesion. This multilayer film possessed a contact-killing effect reducing ~90% the viability of Gram-positive *S. aureus* and Gram-negative *Pseudomonas aeruginosa* (*P. aeruginosa*) strains after 48 h contact. Moreover, Wharton’s jelly (WJ-SCs), dental pulp (DPSCs), and bone marrow (BM–MSCs)-derived stromal cells were not only able to produce immunomodulatory mediators but also release antibacterial and antibiofilm agents effective against *S. aureus* and *P. aeruginosa* strains [100]. In the same line, HA was also alternatively deposited with polyarginine (PAR) leading to antimicrobial multilayer films. By varying the chain lengths of PAH (10, 30, 100, and 200 residues), it was demonstrated that only films built up with poly(arginine) composed of 30 residues (PAR30) acquired an efficient antimicrobial activity against Gram-positive and Gram-negative pathogenic bacteria associated with medical devices infections. Mobile PAR30 chains diffused toward the multilayer film interface inhibited bacteria growth through a contact-killing mechanism (Figure 6a) [101]. It has been also proven that PAR/HA films not only showed an antimicrobial activity against *Staphylococcus aureus* for 24 h, but also reduced the inflammatory cytokines released by human primary macrophages avoiding chronic inflammatory after implantation. Moreover, the incorporation of catestatin (CAT), a natural antimicrobial peptide, leaded to PAR/HA + CAT films with antimicrobial properties against yeast and fungi as well [102].

Recently, collagen (COL)/tannic acid (TA) multilayer films have been developed using two different buffers, acetate or citrate at pH 4. TA/COL binding is stronger in citrate buffer than in acetate buffer leading to a higher immobilization of TA, a natural polyphenol that inhibits the growth of several bacterial strains, and a granular topography of the resulting films, which also provided a higher surface area for an enhanced local release-killing effect against *Staphylococcus aureus* (Figure 6b) [103].

### 5.2. Platforms for Tissue Regeneration

The spray deposition of PLL and HA layers was further employed to coat porous scaffolds of modified hyaluronic acid to give rise to a multilayer 3D structure, formed by the multilayer PLL/HA film covering the scaffold, which promoted the adhesion and proliferation of keratinocytes creating an epidermal structure that mimicked the natural skin microenvironment for potential use in skin tissue engineering applications [104]. Besides, PLL/HA multilayers were employed as a precursor soft film to support a vesicle layer mimicking the ECM, proving that the structure and the interaction with cells were similar to those of native extracellular vesicles [105]. PLL/HA multilayers were also employed to coat PLGA tubes and encapsulate a bone morphogenetic protein (BMP-2). It was proven that the coated tubes were able to repair a critical size volumetric bone femoral defect in a rat. Moreover, the amount of cortical bone increased with the BMP-2 dose (Figure 7a) [106]. Moreover, further studies in vivo conducted by Mano and co-workers showed that (CHI/ALG/CHI/HA-DN)_100_ multilayer films mimicking the features of the native ECM could be also used for skin wound healing. The results point to a progress of the wound healing process after 21 days, showing a thin epidermal layer and epidermal papillaes in the case of HA–DN films. Additionally, the mastocyte population was also reduced for HA–DN films leading to a highest decrease in inflammation [107]. In this sense, multilayer PLL/PGA films with a biological activity based on a synergy effect of two active compounds, BMP2 and TGFb1 growth factors, were also employed for cartilage and bone differentiation. In vitro studies with mesenchymal stem cells revealed that cells came into contact with the growth factors through the PLL/PGA films due to the degradation by cells and both growth factors, BMP2 and TGFb1, needed to be present simultaneously in the film to drive the embryoid bodies (EBs) to cartilage and bone formation [108]. Apart from growth factors, proteins and enzymes can also be incorporated in multilayer films with PLL as polycation and they also maintained their activity while they were encapsulated [62,109,110]. In the case of proteins, it has been proved that proteins interacted with cells through local degradation of the PLL/PGA film [109,110]. Another approach to use multilayer films in tissue engineering applications is the formation of micro-stratified structures composed of PLL-based multilayers with gel layers in between these. For that purpose, micro-stratified structures composed of PLL/HA layers with alternate alginate gel layers were built up through spray assisted LbL where ALG gel layers were obtained by crosslinking ALG with CaCl_2_ using two different experimental methods, spraying or dipping, which led to microporous gels or homogeneous gels, respectively [111]. This micro-stratified structure was further employed to incorporate fibroblastic 3T3 cells or melanocytic B16-F1 cells into the ALG gel layers. The multilayers PLL/HA had the following functions: act as a separator between two gel layers containing different types of cells, work as reservoirs for biologically active compounds interacting with cells embedded in the adjacent gel layers due to their exponential growing and offer mechanical stability to the gel. This was also studied for other polymer systems with PLL/PGA acting as a multilayer structure between gel layers. It was observed that the bioactivity of the multilayer films originates mainly from the local degradation carried out by the cells and the cellular activity could be tuned as a function of the nature of polymer multilayers and the position of active molecules in the architecture [112]. ALG gels have been also employed as 3D customizable sacrificial microstructures to develop 3D vascularized tissue constructs. For such purpose, ALG tubular gels were LbL coated with CHI and arginine–glycine–aspartic acid (RGD)-grafted ALG multilayers. Then, they were embedded in photo-cross-linkable xanthan gum hydrogels and exposed to a calcium chelating solution leading to perfusable multilayer microchannels. Subsequently, ALG gels were liquefied using ethylenediaminetetraacetic acid (EDTA) generating hollow multilayer microchannels embedded within the xanthan gum hydrogel matrix mimicking 3D vascularized tissue constructs to be employed for the HUVECs cell culture (Figure 7b) [113]. CHI/ALG films have been also employed as scaffolds to produce 3D aggregates of human adipose stem cells (hASCs) [82].

COL has also been combined with other polyanions, HA and ChS, to be employed as coatings of different materials, such as poly(L-lactic acid) [115,116] and polyurethane (PU) [114], to enhanced their biocompatibility for tissue engineering applications. COL membranes have been functionalized with CHI/ALG/calcium phosphate by simultaneous spraying, showing an enhanced proliferation of MSCs with the secretion of cytokines and growth factors that could block bone resorption and favor endothelial cell recruitment. This system improved MSCs regenerative capacity to support bone tissue vascularization and modulate inflammatory infiltrates [117]. Denaturalization of collagen by acid and alkaline processes gives rise to another natural polymer known as gelatin (GL). The assembly of GL and HA has been employed as a biomaterial coating, in this case for polyethylene terephthalate (PET) artificial ligament grafts. Their biocompatibility was proved either in vitro with human dermal fibroblasts (HDF) or in vivo on anterior cruciate ligament reconstruction in rabbits and pigs, demonstrating that these multilayer coatings inhibited inflammatory cell infiltration and promoted new ligament tissue regeneration among the graft fibers enabling their use as coatings for ligament reconstruction applications (Figure 7c) [114]. Micro-stratified structures based on GL/CHS multilayers were used to coat gelatin gels giving rise to double membrane hydrogels with enhanced mechanical properties with respect to interpenetrating polymer networks, reaching elastic modulus close to the extracellular matrix while maintaining the thermo-responsive behavior [118].

### 5.3. Matrixes for Drug Delivery

The possibility of incorporating different therapeutic agents into multilayer films is of vital importance to treat diverse diseases. In this context, LbL films have been employed to load vitamins [119], antitumor therapeutic agents, such as sodium diclofenac, paclitaxel and tamoxifen [70,120], and anti-inflammatory drugs, such as piroxicam (Px), inside films with different number of bilayers [121].

Multilayer CHI/ALG systems have been evaluated as permeation membranes of model drugs (i.e., fluorescent dextrans), where the dextran molecular weight plays a key role on the diffusion rate throughout the multilayer membrane modulating the sustained drug release over the time (Figure 8a) [47]. For antitumor applications, latanoprost, an antiglaucoma ophthalmic drug, was loaded into the multilayer structure to study their effect in vitro and in vivo proving that this pad reduced the intraocular pressure (IOP) in patients with glaucoma disease [122]. Recently, tamoxifen (TMX), a specific drug against breast cancer, was embedded within CHI/ALG multilayer films in different intermediate positions leading to tunable sustained release platforms to be employed as localized drug delivery patches (Figure 8b). The results showed a selective effect of tamoxifen released from multilayer CHI/ALG films against human breast cancer cell viability (MCF-7), whereas fibroblasts’ viability remained unaltered [120]. For another therapeutic application, adenosine deaminase inhibitor was loaded in an intermediate position of CHI/ALG films to study its release, which was attributed to a diffusion-controlled mechanism. This free-standing nanofilm could act as a nanopatch for targeted anti-inflammatory drug delivery to treat localized pathologies as inflammatory bowel disease [123]. CHI/ALG multilayers have been also employed to coat bovine serum albumin (BSA)-loaded gelatin core gels in order to modulate the drug delivery [124]. Apart from ALG, CHI has been also combined with HA to build up drug delivery platforms. In this regard, antitumor drugs, such as sodium diclofenac and paclitaxel, were incorporated into CHI/HA films giving rise to a reduction of human colonic adenocarcinoma HT29 cell viability over three days in contact with these loaded films [70]. Furthermore, these CHI/HA multilayers were evaluated to act as localized drug delivery systems promoting the artery healing process, incorporating L-arginine into the multilayer structure. Results showed that these films improved their protective effect against platelet adhesion, as compared to arteries protected by a film without L-arginine [86]. There are different factors, such as the number of bilayers, pH of the drug loading solution, and the ionic strength of solution, which influence the drug-loading capacity into multilayer films by diffusion. Therefore, in a different approach, multilayer CHI/HA films were immersed in myoglobin (Mb) solution at pH 5.0 giving rise to a gradual load of Mb into the films. Thicker films could load more Mb and the incorporated Mb took longer time to reach the equilibrium. Positively charged Mb at pH 5.0 demonstrated more loading amount than negatively charged Mb at pH 9.0 and neutral Mb at pH 7.0, showing that the main driving force for the bulk loading of Mb was most probably the electrostatic interaction between oppositely charged Mb in solution and HA in the films, while other interactions such as hydrogen bonding and hydrophobic interaction may also play an important role. The ionic strength or the concentration of NaCl in the Mb loading solution also influenced the loading behavior. As the ionic strength of Mb loading solution increased, the quantity of Mb loaded increased and the corresponding loading time decreased [125]. LbL assemblies of CHI and heparin (HEP) were prepared to be used as reservoirs for the long-term storage (up to 9 months) of acidic and basic fibroblast growth factors, aFGFs, and bFGFs, respectively. The growth factors released, aFGF and bFGF, were modulated by the architecture and composition of the multilayer structure. Cell culture assays, performed in contact with NIH-3T3 fibroblast cells, demonstrated the enhanced proliferation of NIH-3T3 fibroblasts when aFGF and bFGF were incorporated into the (HEP/CHI)_n_ multilayer structure (Figure 8c) [126]. Multilayer CHI/ALG films have been employed to load antibodies with binding activity of the antigen to the immobilized antibody which can be tuned by pH control converting these films into good candidates as sensitive immunosensors [127].

Another important natural polycation employed is PLL, which has been combined with HA leading to multilayer PLL/HA films. These films were loaded with antitumor drugs and showed a decrease in human colonic adenocarcinoma HT29 cell viability over three days [70]. In another system, PLL/HA films were loaded with anti-inflammatory cytokine (IL-4) to decrease the implant failure due to the immune reactions (Figure 8d). IL-4 release stimulated the differentiation of primary human monocytes, seeded on top, into pro-healing macrophages phenotype producing anti-inflammatory cytokines (IL1-RA and CCL18) and decreasing the pro-inflammatory cytokines secreted (IL-12, TNF-α, and IL-1β) [128]. PLL/HA films have also been used as carriers for delivering recombinant human bone morphogenetic protein 2 (BMP-2). PLL/HA films could retain high and tunable quantities of BMP-2 and deliver it to C2C12 myoblast cells inducing their differentiation in osteoblasts. Moreover, BMP-2 activity was maintained when trapped in the biopolymeric film in hydrated and dried conditions [129]. In the same line, PLL was combined with PGA to build up PLL/PGA films which were loaded with an anti-inflammatory drug. These films provided an anti-inflammatory activity controlled over the time by adjusting the multilayer architecture [121]. Besides, different kinds of growth factors, such as HIV-1 TAT, basic fibroblastic factor (bFGF), and alpha-melanocyte stimulating hormone (α-MSH), have been embedded into PLL/HA, PLL/CHS and PLL/PGA multilayer films. These growth factors maintained their long-time activity when they were embedded into the multilayer structure and after crossing the multilayer membrane, whereas their short-time activity depended on their integration depth [78,108,130]. HA-based films have also been employed as supports to induce the growth of photo-responsive alginate nanogels used as folic acid reservoirs. The exposure of these supported nanogels to LED light induced their photo-triggering and subsequent folic acid release [119].

### 5.4. LbL Nanocomposite Films as Platforms for Thermal Therapies

Almost any type of charged species, including inorganic molecular clusters, nanoparticles, nanotubes and nanowires, can be successfully used as components to prepare LbL nanocomposite films. Within this context, the incorporation of different thermos-responsive agents (i.e., gold or magnetic nanoparticles) within the multilayer structure can enlarge the range of application of polysaccharide membranes to be employed in hyperthermia therapies to treat cancerous lesions. In these cases, a precise control of the thermal treatment is of vital importance to avoid irreparable damages in healthy tissues. Mattoli and co-workers fabricated freestanding ultrathin and mucoadhesive composite CHI/ALG films by embedding gold nanoparticles (Au-NPs) within the multilayer structure leading to thermonanofilms (TNFs), which were used for controlled photo-thermal ablation of tissues through laser irradiation. In vitro assays were carried out using human neuroblastoma cancer cells (SH-SY5Y) and results showed the TNF-mediated thermal ablation of SH-SY5Y cells (Figure 9a). Moreover, ex-vivo assays were performed with chicken breast tissue and results exhibited a localized vaporization and carbonization of the muscular tissue in contact with a TNF produced by the laser-induced temperature increase (ΔT > 50) [131]. Apart from Au-NPs, magnetite (Fe_3_O_4_) nanoparticles can be used as heating agents for magnetic hyperthermia therapies. The incorporation of Fe_3_O_4_ NPs within CHI/ALG multilayer structure gave rise to thermomagnetic films (TMFs) [64,132]. In presence of an alternating magnetic field (AMF) (f = 180 kHz; H = 35 kA.m^−1^), TMFs showed a temperature increase modulated by the number of Fe_3_O_4_ NP layers deposited along the multilayer film, from 6 °C for 80 NPs layers up to 12 °C for 160 NPs layers within the first 5 min. The employment of these TMFs as heating substrates for magnetic hyperthermia was tested in vitro with SH-SY5Y cells. The exogenous magnetic hyperthermia after application of two AMF cycles (30 min each) resulted in a 85% reduction of SH-SY5Y cell viability, opening the door to treatments of small, superficial, and local tumor lesions (Figure 9b) [132].

A summary of all works referenced in this review article is collected in Table 1.

## 6. Conclusions and Future Perspectives

This article presented a comprehensive overview of layer-by-layer assembly of natural polymers for biomedical applications being polysaccharides and polyamino acids, the most employed for the LbL assembly of multilayer polymer films. In particular, chitosan (CHI) and poly (L-lysine) (PLL) are commonly employed as polycations, whereas alginate (ALG) and hyaluronic acid (HA) are the most studied polyanions as seen in Table 1 where a summary of all works referenced in this review article is provided. In Table 1, the information is classified as a function of the polycation (i.e., PLL, CHI, COL and GL) and the polyanion employed for the assembly and the biomedical applications of the multilayer assemblies obtained. PLL/HA, CHI/HA, and CHI/ALG films have been widely studied as substrates for cell adhesion, where cell adhesive properties are not only influenced by the biological properties of the biopolymer system but also by the mechanical properties. The blood-compatibility properties of CHI/HEP and COL/HEP films made them ideal candidates to be employed as coatings for different implants such as titanium implants or stainless-steel stents for tissue engineering applications. Moreover, CHI/HA and CHI/HA–DN films exhibited excellent adhesive and anti-inflammatory properties to be used as patches for wound healing applications.

In relation to the different methods of preparation found in the literature for the preparation of multilayer films from natural polymers, it is possible to conclude that even if dipping LbL was the first LbL technique employed and the technique most commonly employed, many works employ spray LbL since its discovery by Ciba Vision. Hence, spray LbL constitutes an advantageous and powerful technique not only for scientific research but also for industrial purposes. Apart from these two LbL techniques, spin coating and, very recently, brush coating have also been employed for building up multilayer polymer films. The influence of several experimental variables (i.e., pH, molar mass, ionic strength, or nature of the polyanion) was also discussed, highlighting their influence on the growth mechanism (linear of exponential) as it was exhibited by each specific polycation/polyanion system which, in turn, controls the final properties of the multilayers.

From our point of view, the employment of natural polymers as building blocks of advanced materials for biomedical applications will continue to increase scientific and technological interest, not only because of their intrinsic characteristics as to their biodegradability and biocompatibility but also because of their versatility to be functionalized to get semi-synthetic polymers. LbL assembly of natural polyelectrolytes constitutes a straightforward methodology to produce polymer multilayer films (self-standing and as coatings of organic and inorganic substrates) under mild reaction conditions as reported in this manuscript. Current trends in the fabrication of LbL materials from natural polymers that exploit the assembly of these polyelectrolytes onto soft colloidal nanostructures (i.e., nanogels, vesicles, liposomes or micelles among others) can be developed as templates for the design of biomaterials for the encapsulation and controlled release of active molecules. In this regard, three-dimensional LbL assembly of natural-based polymers modified and/or functionalized with biological moieties will extend the range of biomedical applications reported herein. The addition of inorganic nanoparticles within LbL multilayer films to yield nanocomposite polymer films constitutes a current strategy for functionalization of multilayer polymer films which has also been addressed in this review. In our opinion, research efforts on the development of nanocomposite multilayers need to be made in order to promote effective interaction between the nanoparticles and the polymer matrix (i.e., nanoparticles functionalization) in order to control the nanoparticle’s degree of dispersion and, hence, the final properties exhibited by the multilayers.

## Figures and Tables

**Figure 1 polymers-13-02254-f001:**
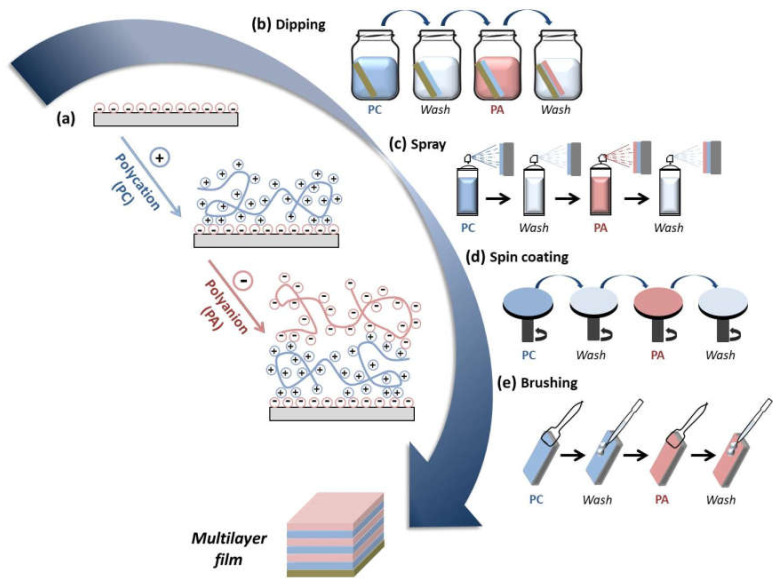
(**a**) Sequential deposition of polycations (PCs) and polyanions (PAs) during the LbL assembly leading to the buildup of a multilayer film; different layer-by-layer (LbL) deposition techniques can be employed for the buildup of multilayer films: (**b**) dipping; (**c**) spray; (**d**) spin coating; (**e**) brushing.

**Figure 2 polymers-13-02254-f002:**
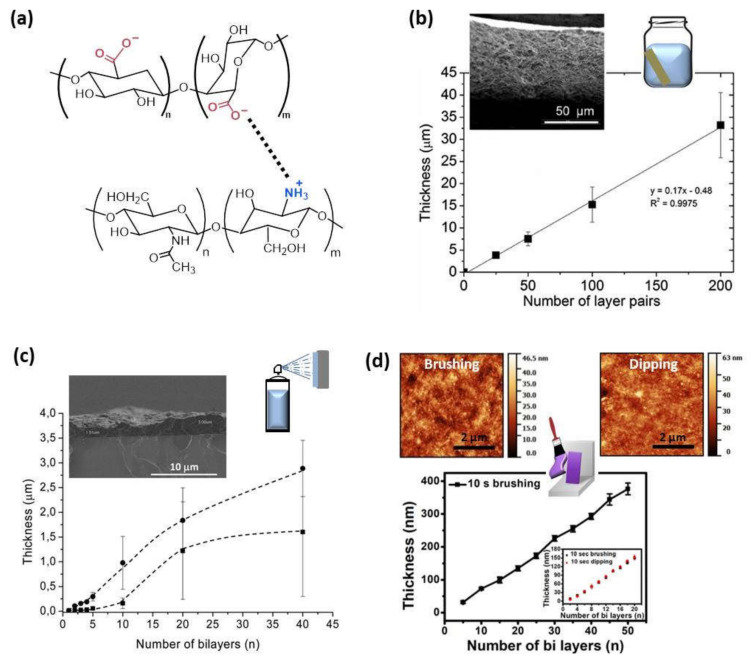
(**a**) Schematic representation of the electrostatic interaction between the protonated amine groups of chitosan and carboxylate groups of alginate; (**b**) Thickness of CHI/ALG films built up by dipping and a cross-sectional image via SEM (Adapted and reprinted with permission from [47]; Copyright 2013 American Chemical Society); (**c**) Evolution of the thickness of (ALG/CHI)_n_ films with different ALG concentrations: 1 mg/mL (■) and 2.5 mg/mL (●) measured by SEM. The inset of the figure is a representative SEM micrograph of a film’s cross-section (Reprinted with permission from [48]; Copyright 2016 American Chemical Society); (**d**) Growth thickness of CHI/ALG films assembled by brushing with a comparison of dipping and brushing methods in the inset and in the topographic AFM images (adapted and reprinted with permission from [33]. Copyright 2018, Kyungtae Park et al. Springer Nature).

**Figure 3 polymers-13-02254-f003:**
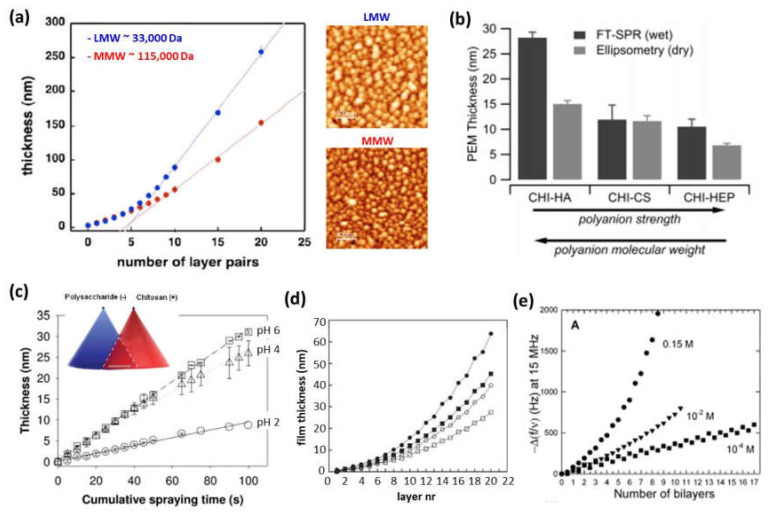
(**a**) Thickness of (CMCHI/CHI)_n_ built-up by spray using LMW chitosan (blue curve) or MMW chitosan (red curve) and AFM topographic images of multilayer films with 20 layer pairs (Adapted and reprinted with permission from [45]; Copyright 2012 Maria A. Witt et al.); (**b**) Thickness, measured by surface plasmon resonance (SPR) and ellipsometry, of CHI-based films assembled in the presence of different polyanions (i.e., HA, CHS, and HEP) (Reprinted with permission from [51]; Copyright 2011 American Chemical Society); (**c**) Thickness of CHI/CHS films built up by simultaneous spraying at a fixed salt concentration of 150 mM NaCl at different pHs: 2 (○), 4 (△), and pH 6 (□) (Adapted and reprinted with permission from [52]; Copyright 2012 American Chemical Society); (**d**) Film thickness after sequential deposition of CHI and HEP at different ionic strengths and pHs: 150 mM NaCl and pH 5.8 (●), 150 mM NaCl and pH 4.2 (■), 30 mM NaCl and pH 5.8 (○), and 30 mM NaCl and pH 4.2 (□) (Adapted and reprinted with permission from [42]; Copyright 2011 American Chemical Society); (**e**) Differences in the QCM frequency shifts at 15 MHz for CHI/HA multilayer films at pH 5 and different NaCl concentrations (0.15 M, 10^−2^ M, and 10^−4^ M NaCl) (Reprinted with permission from [25]. Copyright 2004 American Chemical Society).

**Figure 5 polymers-13-02254-f005:**
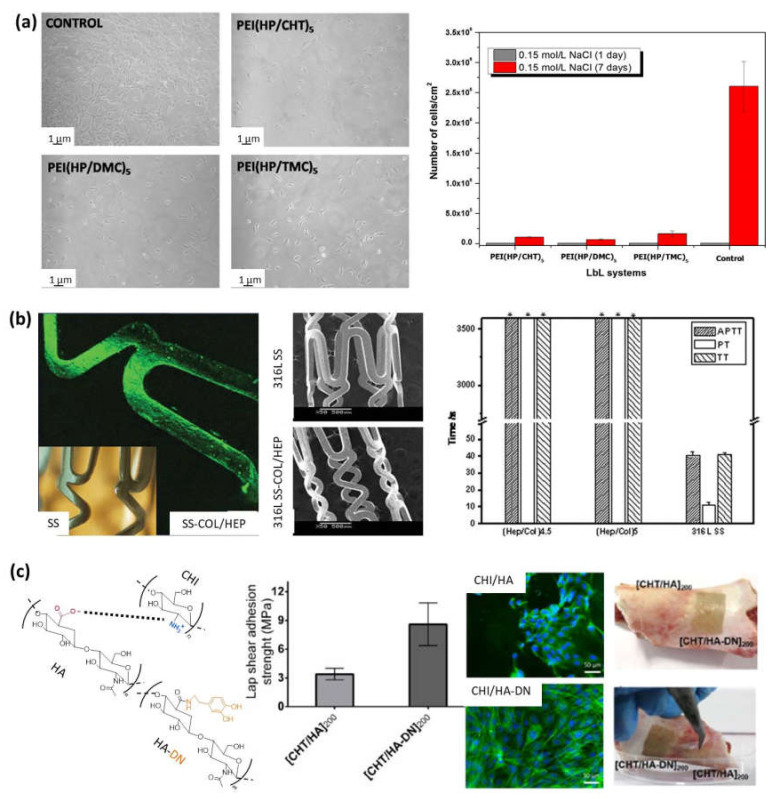
(**a**) NIH/3T3 fibroblasts after 1 day of contact with controls or LbL multilayer films (left), and surface cell density after 1 and 7 days (right) (Adapted and reprinted with permission from [84]; Copyright 2016 Elsevier Inc. All rights reserved); (**b**) Adapted and reprinted with permission from [90] Copyright 2010 Wiley Periodicals, Inc.; (**c**) Schematic representation of the electrostatic interaction between chitosan and hyaluronic acid (left). Adhesive properties of freestanding membranes CHI/HA and CHI/HA-DN (center). Osteopontin immunofluorescence images of MC3T3-EI cells on those membranes, and images of the adhesiveness potential of those freestanding membranes on a clean surface of porcine bone: before (right-top) and after applying a detachment force with tweezers (right) (Adapted and reprinted with permission from [91]. Copyright 2017 MDPI).

**Figure 6 polymers-13-02254-f006:**
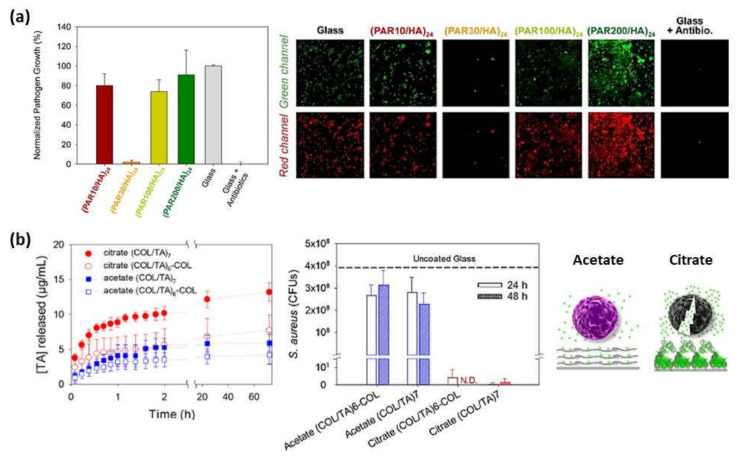
(**a**) Normalized *S. aureus* growth after 24 h contact with (PAR/HA)_24_ multilayer films composed of PAR with a variable number of residues per chain (left) and confocal images (67 × 67 μm^2^) of *S. aureus* after 24 h of incubation on glass substrates, (PAR/HA)_24_ films built with PAR10, PAR30, PAR100, or PAR200 and on a glass slide with antibiotics (right). The green channel corresponds to SYTO 24 labeling all bacteria and the red channel only healthy bacteria through metabolism of CTC (5-cyano-2,3-ditolyl tetrazolium chloride) into an insoluble, red fluorescent formazan. (Reprinted with permission from [101]; Copyright 2016 American Chemical Society); (**b**) Cumulative release profiles of TA from TA/COL films in contact with PBS pH 7.4 at room temperature (left), CFU of *S. aureus* from the supernatant of TA/COL films after 24 and 48 h of contact (N.D. means not detected, no CFUs were observed; dashed line means CFUs were obtained from the bacterial supernatant in contact with an uncoated glass) (center), and schematic representation of TA release-killing toward bacteria (right) (Adapted and reprinted with permission from [103]. Copyright 2020 American Chemical Society).

**Figure 7 polymers-13-02254-f007:**
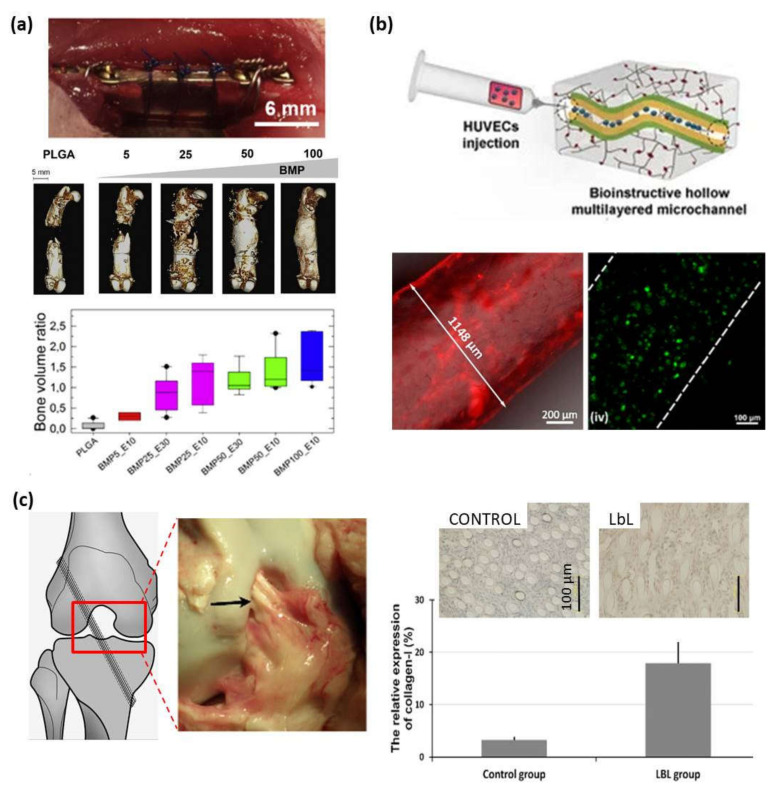
(**a**) Post-operative view of the implanted PLGA tube maintained by suture wires (top), CµCT reconstructions (center) taken at 8 weeks after implantation at different doses of BMP-2 (scale bar = 5 mm) (center) and bone volume ratio at 8 weeks (bottom) (Adapted and reprinted with permission from [106]; Copyright 2016 Michael Bouyer et al., Published by Elsevier Ltd.); (**b**) Schematic representation of bioinstructive (CHI/ALG-RGD)_6_ multilayers over liquefied ALG microchannels embedded in photocross-linkable glycidyl methacrylated xanthan gum (XG-GMA) hydrogels (top). Fluorescence microscopy image of a 3D ALG sacrificial microfiber coated with (RITC-CHI/ALG)_6_ multilayer film (bottom-left), and live/dead confocal micrographs of HUVECs seeded for 3 days in the microchannels (bottom-right) (Reprinted with permission from [113]; Copyright 2021 by Cristiana F. V. Sousa et al. Licensee MDPI); (**c**) PET ligament, surgical schematic diagram of an anterior cruciate ligament (ACL) reconstruction model, and gross graft sample observation 3 months after operation in a porcine model (left). Immunohistochemical staining for collagen type I and relative expression of collagen type I (right) (Adapted and reprinted with permission from [114]. Copyright 2012 Acta Materialia Inc. Published by Elsevier Ltd. All rights reserved).

**Figure 8 polymers-13-02254-f008:**
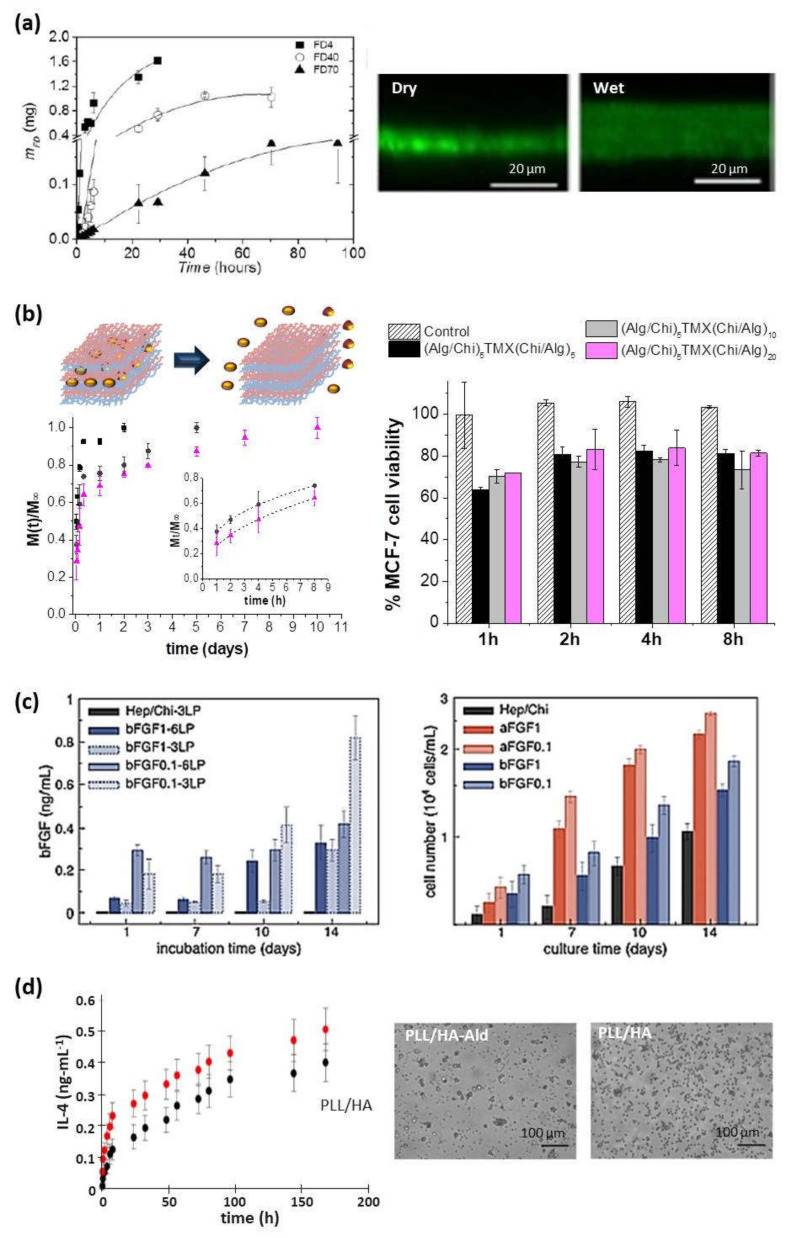
(**a**) Diffusion of FITC-dextran (FD) of different molecular weights (4, 40, and 70 kDa) through a (CHI/ALG)_100_ film (left) and CLSM images (right) (Reprinted with permission from [47]; Copyright 2013 American Chemical Society); (**b**) Schematic representation of drug delivery from multilayer films and cumulative release profiles of tamoxifen (TMX) from (ALG/CHI)_5_TMX(CHI/ALG)_m_ (m = 5 (■), 10 (●), and 20 (▲)) films, where the inset shows the fitting to the Ritger-Peppas model, (left) and percentage of MCF-7 cell viability after tamoxifen (TMX) release from these films comparing with control films without TMX (right) (Adapted and reprinted with permission from [120]; Copyright 2018 Elsevier Ltd.); (**c**) Release profiles of bFGF reservoirs assembled with 3LP or 6LP and from co-solutions with bFGF at 0.1 or 1 µg/mL (left) and NIH/3T3 fibroblasts proliferation on FGF reservoirs with 6LP (right) (Adapted and reprinted with permission from [126]; Copyright 2015 Elsevier B.V.); (**d**) Kinetic of IL-4 release from PLL/HA and PLL/HA-Ald films in PBS (left) and phase contrast images (right) of monocytes seeded on the films after 6 days in presence of IL-4. Scale bar = 100 μm. (Adapted and reprinted with permission from [128]. Copyright 2016 American Chemical Society).

**Figure 9 polymers-13-02254-f009:**
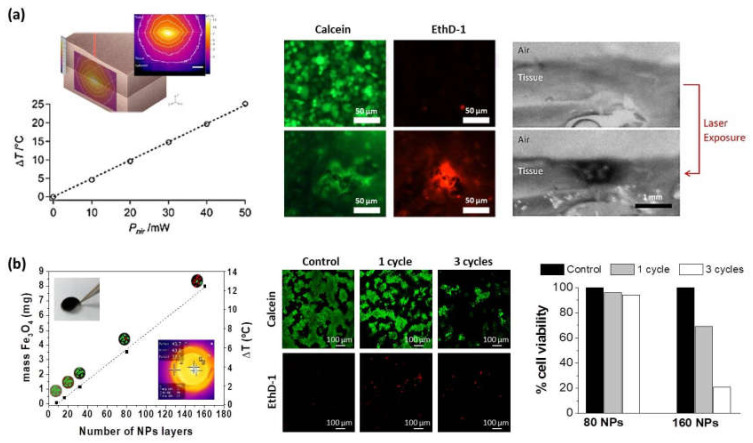
(**a**) Maximum temperature increase dependence on nominal laser power for a thermofilm TNF10x with the inset showing the TNF-mediated thermalization of chicken breast tissue upon laser irradiation. Cell viability assay: calcein and EthD-1, signal before and after laser exposure (λ = 785 nm, 50 mW, 150 s) of TNF10x in contact with human neuroblastoma cells (Adapted and reprinted with permission from [131]; Copyright 2014 American Chemical Society); (**b**) Determination of Fe_3_O_4_ content and temperature increase as a function of the number of NP layers deposited in the presence of an alternating magnetic field (AMF) (left). (ALG/CHI)_40_(NPs/CHI)_160_ films subjected to 0 cycles of MHT (control), 1 cycle of MHT (30 min), and 3 cycles of MHT (30 min, 5 h waiting time between cycles). Note that cells are labeled with calcein (live cells in green) and EthD-1 (dead cells in red). Determination of % cell viability obtained from the images corresponding to live cells stained in green (Adapted and reprinted with permission from [132]. Copyright 2017 The Royal Society of Chemistry).

**Table 1 polymers-13-02254-t001:** Summary of the multilayer films based on natural polymers with PLL, CHI, COL, and GL as polycations and several polyanions, type of LbL assembly technique employed, and potential biomedical application.

Polycation	Polyanion	Biomedical Applications	References
PLL	ALG	Wound healing Bioactive coatings tissue engineering	[16] [95]
	HA	Cell adhesive membranes Tissue engineering membranes Antitumor pads Drug delivery Reservoirs for growth factors	[61,69,70,71,72,73,74,78] [104,105,106,111] [70] [4,128,129] [78,108,128,129,130]
	CHS	Cell adhesive membranes Reservoirs for growth factors	[78] [78]
	PGA	Cell adhesive membranes Tissue engineering membranes Drug delivery Reservoirs for growth factors	[40,83] [108,109,110,112] [108,109,110] [108]
CHI	HA	Cell adhesive membranes Bioactive coatings tissue engineering Tissue engineering membranes Wound healing Antimicrobial coatings Antitumor pads Drug delivery	[54,75,81] [81,94] [86,91] [107] [40,99,100] [70] [86,125]
	ALG	Cell adhesive membranes Bioactive coatings tissue engineering Tissue engineering membranes Sensitive immunosensors Antimicrobial coatings Thermal patches Drug delivery Antitumor pads	[47,58,70,76,77,79,82] [81,94] [113,117] [127] [96] [131,132] [47,123,124] [120,122]
	HEP	Anticoagulant coatings Anti-inflammatory coatings Wound healing Bioactive coatings tissue engineering Antimicrobial coatings Drug delivery Reservoirs for growth factors	[8,24,85] [8,84] [8] [92,93] [97,98] [126] [126]
	Dex	Anticoagulant coatings	[24]
	CHS	Cell adhesive membranes	[65]
COL	HA	Cell adhesive membranes Bioactive coatings tissue engineering	[43] [115]
	HEP	Cell adhesive membranes Anticoagulant coatings Bioactive coatings tissue engineering	[44] [87,88,89] [90]
	CHS	Cell adhesive membranes Bioactive coatings tissue engineering	[44] [116]
	ALG	Cell adhesive membranes	[9]
	TA	Antimicrobial coatings	[103]
GL	HA	Bioactive coatings tissue engineering	[114]
	CHS	Tissue engineering membranes	[118]

## Data Availability

There are no data associated with this publication.

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
