# Peer review of "Polyelectrolyte Multilayer Films Based on Natural Polymers: From Fundamentals to Bio-Applications"

_polymers, 2021, doi:10.3390/polym13142254_

Round 1
Reviewer 1 Report
The Review prepared by Authors seems to be an interesting and well-organized work. It constitutes a range of information on the selected research topic. In general, the work is proper and worth noticing but some elements need to be improved. All suggestions are given below.
- Introduction of the paper needs to be significantly improved to be consistent and fluent. For example, in the second paragraph some biopolymers have been described in few sentences one after the other but there is no flow in this description. Next, the third paragraph begins with the sentence “Another interesting property…” but what does the word “another” refer to?
- Figure 2. should be divided into several smaller ones – now subfigures are too small and therefore slightly difficult to see. The same applies to Figure 4. and Figure 8.
- The paper should be supplemented with additional section containing all abbreviations with their explanations.
- From the editorial viewpoint, the notation of citations in the paper should be corrected. The citation in brackets needs to be placed before the dot in a sentence (not vice versa as it is in the article).
- Section Conclusions: more attention should be paid to the specific conclusions and the perspectives for the future. This section is too general.
- Section References is not consistent – some references contain the whole journal names and some of them contain their abbreviations. It needs to be unified.
- Language of the paper needs to be re-checked.
Author Response
We greatly appreciate the comments the comments from reviewer #1
- Introduction of the paper needs to be significantly improved to be consistent and fluent. For example, in the second paragraph some biopolymers have been described in few sentences one after the other but there is no flow in this description. Next, the third paragraph begins with the sentence “Another interesting property…” but what does the word “another” refer to?
ANSWER. We thank the referee for the suggestion. The introduction of the paper has been reorganized for improvement of consistency in between paragraphs.
- Figure 2. should be divided into several smaller ones – now subfigures are too small and therefore slightly difficult to see. The same applies to Figure 4. and Figure 8.
ANSWER. Figures 2, 4 and 8 have been divided in smaller ones to increase the size of every subfigure.
- The paper should be supplemented with additional section containing all abbreviations with their explanations.
ANSWER. The abbreviations section has been included at the end of the manuscript.
- From the editorial viewpoint, the notation of citations in the paper should be corrected. The citation in brackets needs to be placed before the dot in a sentence (not vice versa as it is in the article).
ANSWER. The notation of citations has been modified accordingly.
- Section Conclusions: more attention should be paid to the specific conclusions and the perspectives for the future. This section is too general.
ANSWER. In the revised manuscript we have increased the information about the perspectives of the field for the future. The specific conclusions extracted from the review of the literature have also been rephrased to be less general.
- Section References is not consistent – some references contain the whole journal names and some of them contain their abbreviations. It needs to be unified.
ANSWER. The section of references has been unified with abbreviated journal names.
- Language of the paper needs to be re-checked.
ANSWER. Language of the paper has been reviewed in the revised manuscript through the MDPI author services.
Reviewer 2 Report
The review article (Manuscript Number polymers-1289190) describes the crucial advances concerning multilayer assembly of natural polymers employing the most used layer-by-layer (LbL) techniques, like dipping, spray and spin coating, leading to multilayer polymer structures and the influence of several variables, such as pH, molar mass, method of preparation, in this LbL assembly process.
As the authors stated, this interesting review article gives a systematic overview of synthetic approaches of the novel polyelectrolyte multilayer films based on natural polymers, briefly introduces their use in sample pretreatment prior to bio-applications, and provides a perspective for future research. These multilayer biopolymer films can be successfully used as platforms for tissue engineering, drug delivery, and thermal therapies.
This review article includes a balanced, comprehensive, and critical view of the research area. On the whole, the manuscript is fairly well-written and logically arranged. The overall originality of the review concept used here is high. The presented results are informative, and the discussion is clear. I think that this paper can be published as-is.
Author Response
We greatly appreciate the very positive comments from reviewer 2 that considered that the manuscript can be published as it is.